# Piston Error Measurement for Segmented Telescopes with an Artificial Neural Network

**DOI:** 10.3390/s21103364

**Published:** 2021-05-12

**Authors:** Dan Yue, Yihao He, Yushuang Li

**Affiliations:** College of Science, Changchun University of Science and Technology, Changchun 130022, China; 2019100133@mails.cust.edu.cn (Y.H.); 2020100186@mails.cust.edu.cn (Y.L.)

**Keywords:** piston error detection, segmented telescope, BP artificial neural network, modulation transfer function

## Abstract

A piston error detection method is proposed based on the broadband intensity distribution on the image plane using a back-propagation (BP) artificial neural network. By setting a mask with a sparse circular clear multi-subaperture configuration in the exit pupil plane of a segmented telescope to fragment the pupil, the relation between the piston error of segments and amplitude of the modulation transfer function (MTF) sidelobes is strictly derived according to the Fourier optics principle. Then the BP artificial neural network is utilized to establish the mapping relation between them, where the amplitudes of the MTF sidelobes directly calculated from theoretical relationship and the introduced piston errors are used as inputs and outputs respectively to train the network. With the well trained-network, the piston errors are measured to a good precision using one in-focused broadband image without defocus division as input, and the capture range achieving the coherence length of the broadband light is available. Adequate simulations demonstrate the effectiveness and accuracy of the proposed method; the results show that the trained network has high measurement accuracy, wide detection range, quite good noise immunity and generalization ability. This method provides a feasible and easily implemented way to measure piston error and can simultaneously detect the multiple piston errors of the entire aperture of the segmented telescope.

## 1. Introduction

To fulfill the demands of increasing space exploration, the aperture diameter of the telescope is getting larger and larger [1]. However, the diameter is limited by current technology to about 10 m for ground-based telescopes and a few meters for space-based telescopes because of volume and mass considerations. The segmented and deployable primary mirror was proposed to address this problem, which can effectively reduce the mass of the main mirror and the difficulty of manufacturing and transportation [2]. Ground-based telescopes that have adopted the segmented primary mirror include the already existing 10 m aperture Keck telescopes [3,4] made of 36 segments, the 42 m diameter European Extremely Large Telescope (E-ELT) [5] made of 984 segments and the Thirty Meter Telescope (TMT) made of 492 segments with a 30 m diameter [6], which are planned in the near future. Space segmented telescope projects aim at astronomical or earth observations, such as the 6.5 m James Webb Space Telescope (JWST) [7] composed of 18 segments, to be launched very soon, and the 9.2 m Advanced Technology Large Aperture Space Telescope (ATLAST) [8] composed of 36 segments for a next-generation flagship astrophysics mission to study the universe.

However, segmented-mirror technology introduced new problems. The optical path difference (OPD) between segments, which can be described by the first three Zernike polynomials (piston and tip-tilt), must be reduced to a small fraction of the wavelength to achieve a high-resolution equivalent to that of a monolithic mirror [9]. The crucial point is correcting the piston errors, which cannot be directly detected by traditional wavefront sensors like Shack–Hartmann. The research shows that, when the piston error between segments is reduced to about λ/20, the angle resolution is 1.22λ/D, where D is the diameter of the primary mirror and λ is the observation wavelength. If the piston error is larger than λ, the angle resolution decreases to 1.22λ/d, where d is the diameter of the sub-mirror. For the JWST, which is composed of 18 1.32 m sub-mirrors, the angular resolution difference can reach five times. Thus, in order to guarantee the imaging quality of the segmented telescope, the piston error between the segments must be accurately measured, and the measurement accuracy should be 30~40 nm at least. To achieve this, the detection method of co-phasing the piston error should meet the requirements of large range and high precision.

Currently, many piston error detection methods have been successfully applied to the segmented telescope. The modified Shack–Hartmann wavefront detection method has been successfully used on Keck telescopes. It measures the piston error by installing a Hartmann micro-lens array at the exit pupil. The reliable capture range of the narrow Shack–Hartmann method [10] is ±λ/4 while the measurement accuracy is as high as 6 nm. The broadband Shack–Hartmann algorithm [11] has a wider measurement range and generally reaches 30 μm, while the measurement accuracy is around 30 nm. The quadrilateral pyramid detection method [12,13], which utilizes a quadrangular pyramid mirror, relay lens and CCD camera, can achieve several tens of nanometers’ detection accuracy in the ±λ/2 detection range. The dispersed fringe sensor (DFS) [14,15] is a new piston error detection technique proposed for the next-generation space telescope. The DFS is composed of a prism, a micro-lens array and a CCD camera, which can quickly and accurately detect the piston error from a few microns to tens of microns. All above methods need optical devices which add to the detection hardware complexity. Focal-plane wavefront sensing is an elegant solution to measure the OPD since this wavefront sensor is included in the main imaging detector, simplifying the hardware and minimizing differential paths. The classic focal-plane detection methods, such as phase diversity [16,17,18] and phase retrieval [19,20], have high measurement accuracy but narrow detection range and are very time-consuming. In 2015, Simar [21] found out that the modulation transfer function (MTF) amplitude part of the optical transfer function (OTF) has a relationship with the piston error based on the coherence measurement of a star image when using broadband input light, using Gaussian function to fit the relationship between the MTF and the piston error. The capture range of this method is close to the coherence length of the input light, but its accuracy decreases sharply when approaching the extremes of capture range. Junlun Jiang et al. [22] utilized piecewise quartic polynomial function to fit the relationship; the accuracy of this method is higher than that of Gaussian fitting but the complexity of curve fitting is increased since different mathematical expressions are needed in different intervals of the capture range.

This paper proposes a different method to detect the piston error based on analyzing the broadband intensity distribution on the image plane of a star image with a back-propagation (BP) artificial neural network. By attaching a mask with a sparse multi-subaperture configuration in the exit pupil plane of the segmented telescope, we can get the non-redundant MTF sidelobes distribution. Then, based on Fourier optics principle [23], the theoretical relation between peak heights of MTF sidelobes and piston error is rigorously derived, and we have obtained a more universally applicable and accurate theoretical conclusion than the work presented in paper [22], which clearly indicates that the peak height of the MTF sidelobe is only related to the number of sub-pupils, input wavelength and the piston error between segments. Instead of fitting the relation by Gaussian function or two piecewise quartic polynomials, we utilize the BP artificial neural network to establish the nonlinear mapping relationship between the piston error and the amplitudes of the MTF’s surrounding peaks. The BP artificial neural network [24,25] is simple in structure and very convenient to train. Here, the peak heights of MTF sidelobes directly calculated from the obtained theoretical formula served as the inputs and the introduced piston errors served as outputs, respectively, to train the artificial neural network. Once well trained, it can estimate the piston error to a good precision with high efficiency and robustness using the PSF image collected from the optics imaging system as input. The detection accuracy is both higher than Gaussian function fitting and piecewise quartic polynomial function fitting, and the implementation complexity is lower than piecewise quartic polynomial function fitting since there is no need to build different computation models in different intervals of the capture range. The capture range is close to the coherence length of the input light; thus the piston error detection no longer needs to be divided into coarse and fine regimes which involve separate dedicated hardware solutions. The influence of the CCD camera noise and the existence of tip-tilt aberrations on the accuracy of piston error detection are discussed. The results indicate that the proposed piston error detection algorithm has very good noise immunity, but the existence of tip-tilt aberrations has a great influence on the detection accuracy. Besides, the generalization ability of the network is also discussed by changing the F-number (F^#^) of the imaging system to generate different intensity images as testing sets to examine the trained network. The results show that our network has quite good generalization ability, since the datasets used to train the network are the same when the optical systems have the same number of sub-pupils and working wavelength. Moreover, by matching up the MTF sidelobes with their associated sub-pupils, multiple piston error measurements of the whole aperture can be implemented simultaneously by one detection of a single broadband image. 

This paper is structured as follows. In Section 2, we derive the theoretical relation between the peak heights of MTF sidelobes and piston error, and verify the correctness of the theoretical relation by MATLAB simulation. Section 3 describes how to use the BP artificial neural network to establish the nonlinear mapping relationship between the amplitudes of the MTF sidelobes and the piston error, and how to implement the established network to detect piston errors. Adequate simulation validations and discussions on the proposed approach are presented in Section 4. In Section 5, we conclude the paper.

## 2. Theoretical Relation between the Piston Error and Amplitudes of MTF Sidelobes

In this part, we mainly focus on establishing the theoretical relationship between the amplitudes of the MTF sidelobes and the piston error based on the Fourier optics principle at first. Then, we verify the correctness of the established theoretical relation through MATLAB simulation.

### 2.1. Establishing the Theoretical Relationship Based on Fourier Optics Principle

In order to simplify the theoretical derivation process, we used a primary mirror composed of two hexagonal segments as the optics system model shown in Figure 1. In order to separate the sidelobes of the MTF from the main peak, a mask with two circles was set on the exit-pupil plane of the primary mirror to fragment the pupil. If p is the piston error between the two segments, the generalized pupil function (GPF) can be written as:(1)$$G(x,y)=A(x,y)\left[circ\left(\frac{x-b/2,y}{d/2}\right)\cdot e^{i\phi_{1}}+circ\left(\frac{x+b/2,y}{d/2}\right)\cdot e^{i\phi_{2}}\right],$$
where (x,y) is the coordinate vector of the pupil plane, b is the distance between the center of the two circle pupils on the mask, d is the diameter of the circle pupil, λ is the observation wavelength, circ() stands for circle function, the phase difference between the two segments is $\Delta \phi =\phi_{1}-\phi_{2}=\frac{2\pi }{\lambda }2p$, here we set $\phi_{1}=\frac{2\pi }{\lambda }p$ and $\phi_{2}=-\frac{2\pi }{\lambda }p$ to facilitate the following calculations. A(x,y) is the binary shape function of the hexagon segment which is shown as:(2)$$A(x,y)=\left\{ \begin{array}{l} 1\;\;\;\;\;\;\;\;\;\;\;\;\;\;\;\;\;inside\;\;\;the\;\;pupil\\ 0\;\;\;\;\;\;\;\;\;\;\;\;\;\;\;\;\;others \end{array}\right..$$

We performed Fourier transform for the GPF, and based on the properties of Fourier transform we can derive the following equation:(3)$$ℑ\left\{G(x,y)\right\}=\frac{d}{2}\left[\frac{J_{1}(\pi d\sqrt{u^{2}+v^{2}})}{\sqrt{u^{2}+v^{2}}}\right]\cdot \left(e^{i(\frac{\pi }{\lambda }2p-\pi ub)}+e^{-i(\frac{\pi }{\lambda }2p-\pi ub)}\right),$$
where (u,v) is the coordinate vector of the image plane, ℑ{} stands for Fourier transform operation, $J_{1}\left\{\right\}$ is first order Bessel function. 

Based on the Fourier optics, the point spread function (PSF) of the system is the squared modulus of the Fourier transform of the GPF which is given by: (4)$$\begin{array}{l} PSF\left(u,v\right)={\left|ℑ\left\{G(x,y)\right\}\right|}^{2}={\left(d/2\right)}^{2}{\left|\left[\frac{J_{1}(\pi d\sqrt{u^{2}+v^{2}})}{\sqrt{u^{2}+v^{2}}}\right]\cdot \left(e^{i(\frac{\pi }{\lambda }2p-\pi ub)}+e^{-i(\frac{\pi }{\lambda }2p-\pi ub)}\right)\right|}^{2}\\ \;\;\;\;\;\;\;\;\;\;\;\;\;\;\;\;\;\;\;\;\;\;\;\;\;\;\;\;\;\;\;\;\;\;\;\;\;\;\;\;=2\frac{{\left(d/2\right)}^{2}J_{1}{}^{2}(\pi d\sqrt{u^{2}+v^{2}})}{u^{2}+v^{2}}\left[1+\cos\left(\frac{2\pi }{\lambda }2p-2\pi ub\right)\right]. \end{array}$$

From Equation (4), we can see that the PSF includes two parts: the diffraction part and the interference part. $\frac{{\left(d/2\right)}^{2}J_{1}{}^{2}(\pi d\sqrt{u^{2}+v^{2}})}{u^{2}+v^{2}}$ is the diffraction intensity of the single circle aperture; the coefficient 2 means the diffraction part is simple superposition formed by the two single circle apertures’ diffraction. $\left[1+\cos\left(\frac{2\pi }{\lambda }2p-2\pi ub\right)\right]$ is the interference part which is coherent superposition formed by the sub-waves sampled by the two single circle apertures. As we can see, the piston error is included in the interference part. When the observed target is a point source, the image obtained on the focal plane can be simplified as the PSF of the system, thus the piston error between segments is included in the CCD captured image. According to the interference principle, the interference factor will disappear when the piston error exceeds the coherence length of the light used in the optical imaging system, then the intensity distribution on the image plane becomes the simple superposition of the two sub-pupils’ diffraction. Hence, the capture range of the piston error is limited by the coherence length of the input light.

Based on Fourier optics, the OTF is the 2D Fourier transform of the PSF shown as:(5)$$OTF(f_{x},f_{y})=ℑ\left\{PSF\left(u,v\right)\right\},$$
where $\left(f_{x},f_{y}\right)$ is the spatial frequency in the x and y directions, respectively. According to the OTF calculation method of the diffraction limited system, the expression for the OTF of the segmented system can be given by:(6)$$OTF(f_{x},f_{y})=2OTF_{sub}(f_{x},f_{y})+OTF_{sub}(f_{x}+\frac{b}{\lambda f},f_{y})e^{-i\frac{2\pi }{\lambda }2p}+OTF_{sub}(f_{x}-\frac{b}{\lambda f},f_{y})e^{i\frac{2\pi }{\lambda }2p},$$
where $OTF_{sub}(f_{x},f_{y})$ is the OTF of a single circle aperture diffraction system, which is given by:(7)$$OTF_{sub}(f_{x},f_{y})=\left\{ \begin{array}{l} \frac{2}{\pi }\left[\arccos\left(\frac{\rho }{2\rho_{0}}\right)-\frac{\rho }{2\rho_{0}}\sqrt{1-{\left(\frac{\rho }{2\rho_{0}}\right)}^{2}}\right],\;\;\;\;\;\;\;\;\;\;\;\;\;\;\;\;\rho \leq 2\rho_{0}\;\;\;\;\;\;\;\;\;\;\\ 0\;\;\;\;\;\;\;\;\;\;\;\;\;\;\;\;\;\;\;\;\;\;\;\;\;\;\;\;\;\;\;\;\;,others \end{array}\right.$$
where $\rho =\sqrt{f_{x}^{2}+f_{y}^{2}}$ is the radial coordinate on the frequency plane, $\rho_{0}=\frac{d}{2\lambda f}$ is the system cut-off frequency, f is focal length of the imaging lens. 

Equation (6) shows that the OTF of the segmented telescope composed of two hexagonal segments with mask includes three parts: the central part and two sidelobes. The central part’s spatial frequency is (0,0) and its value is the maximum modules of the sidelobe without piston error multiplied by the number of sub-pupils. Since the OTF is normalized, the value of the central part is always 1, so the maximum height of the sidelobe part without piston error is 1/N, where N is the number of sub-pupils, and N=2 is the telescope composed of two segments. The spatial frequencies of the two sidelobes are (±b/λf,0) respectively, and they are symmetrically distributed about the central part. Based on the Fourier optics, we can get the MTF of the system by doing a modulus operation for Equation (6), which is shown as:(8)$$\begin{array}{l} MTF(f_{x},f_{y})=\left|OTF(f_{x},f_{y})\right|\\ \;\;\;\;\;\;\;=\left|2OTF_{sub}(f_{x},f_{y})\right|+\left|OTF_{sub}(f_{x}+\frac{b}{\lambda f},f_{y})e^{-i\frac{2\pi }{\lambda }2p}\right|+\left|OTF_{sub}(f_{x}-\frac{b}{\lambda f},f_{y})e^{i\frac{2\pi }{\lambda }2p}\right|. \end{array}$$

As we can see, only the value of the MTF sidelobe is modulated by the piston error. We take one sidelobe out marked as $MTF_{sidelobe}$, which is shown as:(9)$$MTF_{sidelobe}(f_{x},f_{y})=\left|OTF_{sub}(f_{x}+\frac{b}{\lambda f},f_{y})e^{-i\frac{2\pi }{\lambda }2p}\right|.$$

Then the peak height value of the sidelobe with piston error marked as $MTF_{sidelobe-peak}$ is obtained by:(10)$$MTF_{sidelobe-peak}=\frac{1}{N}\left|e^{-i\frac{2\pi }{\lambda }2p}\right|.$$

From Equation (10), we can see that the peak height of the MTF sidelobe is only related to the number of sub-pupils, input wavelength and piston error. The effective detection range of the piston error is limited for a single wavelength due to the 2π ambiguity. Therefore, we need a broadband input light value to expand the piston error detection range. 

For a broadband input light centered at $\lambda_{0}$ with the bandwidth Δλ, the PSF of the system is the sum of all PSFs at different monochromatic wavelengths which is given by:(11)$$\begin{array}{l} PSF_{broad}(u,v,\lambda )=\int_{\lambda_{0}-\frac{\Delta \lambda }{2}}^{\lambda_{0}+\frac{\Delta \lambda }{2}}PSF(u,v,\lambda )S\left(\lambda \right)d\lambda \\ \;\;\;\;\;\;\;\;\;\;\;\;\;\;\;\;\;\;\;\;\;\;\;\;\;\;\;\;\;=\int_{\lambda_{0}-\frac{\Delta \lambda }{2}}^{\lambda_{0}+\frac{\Delta \lambda }{2}}2{\left(\frac{d}{2}\right)}^{2}\frac{J_{1}{}^{2}(\pi d\sqrt{u^{2}+v^{2}})}{u^{2}+v^{2}}\left[1+\cos\left(\frac{2\pi }{\lambda }2p-2\pi ub\right)\right]S\left(\lambda \right)d\lambda , \end{array}$$
where S(λ) is PSF weight of different wavelengths, assuming S(λ)=1.

Since the integral is difficult to calculate, a differential summation approximation is used. The Δλ is divided into *n* intervals equally, then the PSF can be rewritten as:(12)$$PSF_{broad}(u,v,\lambda )=\frac{\Delta \lambda }{n}\sum_{t=1}^{n}2{\left(\frac{d}{2}\right)}^{2}\frac{J_{1}{}^{2}(\pi d\sqrt{u^{2}+v^{2}})}{u^{2}+v^{2}}\left[1+\cos\left(\frac{2\pi }{\lambda_{t}}2p-2\pi ub\right)\right].$$

By performing Fourier transform for the $PSF_{broad}$, the corresponding OTF of the system in broadband light marked as $OTF_{broad}$ is given by:(13)$$OTF_{broad}(f_{x},f_{y},\lambda )=\frac{\Delta \lambda }{n}\sum_{t=1}^{n}\left[\begin{array}{l} 2OTF_{sub}(f_{x},f_{y})+OTF_{sub}(f_{x}+\frac{b}{\lambda_{t}f},f_{y})e^{-i\frac{2\pi }{\lambda_{t}}2p}+\ldots \\ OTF_{sub}(f_{x}-\frac{b}{\lambda_{t}f},f_{y})e^{i\frac{2\pi }{\lambda_{t}}2p} \end{array}\right].$$

As we know, performing a modulus operation for the OTF, we can get the MTF of the system and the piston error is only related to the MTF sidelobes, thus the MTF sidelobe marked as $MTF_{sidelobe-broad}$ is extracted out from the modulus of the $OTF_{broad}$ and shown as:(14)$$MTF_{sidelobe-broad}(f_{x},f_{y},\lambda )=\frac{\Delta \lambda }{n}\left|\sum_{t=1}^{n}\left[OTF_{sub}(f_{x}+\frac{b}{\lambda_{t}f},f_{y})e^{-i\frac{2\pi }{\lambda_{t}}2p}\right]\right|.$$

In fact, for different wavelengths, the position coordinates of the MTF sidelobes are different. However, the coordinate difference introduced by different wavelengths is very small when the bandwidth is much smaller than the central wavelength [26], so the position coordinate $\left(f_{x}+\frac{b}{\lambda_{t}f},f_{y}\right)$ with different wavelengths can be approximated as $\left(f_{x}+\frac{b}{\lambda_{0}f},f_{y}\right)$. The effect of different wavelengths is mainly reflected in the e-index part. Thus Equation (14) can be rewritten as:(15)$$MTF_{sidelobe-broad}(f_{x},f_{y},\lambda )=\frac{\Delta \lambda }{n}\left|OTF_{sub}(f_{x}+\frac{b}{\lambda_{0}f},f_{y})\right|\left|\sum_{t=1}^{n}e^{-i\frac{2\pi }{\lambda_{t}}2p}\right|.$$

For the broadband input light value, when the $OTF_{broad}$ is normalized, the value of the central peak is 1. From Equation (13) we can see that the peak height of the MTF sidelobe without piston error, namely $\left|OTF_{sub}(f_{x},f_{y})\right|$, is equal to $\frac{1}{N\Delta \lambda }$ (N=2 for Equation (13)), thus the peak height value of the MTF sidelobe with piston error is
(16)$$MTF_{sidelobe-broad-peak}=\frac{1}{nN}\left|\sum_{t=1}^{n}e^{-i\frac{2\pi }{\lambda_{t}}2p}\right|.$$

Based on the above derivation, we obtained a more universally applicable and precise theoretical conclusion shown in Equation (16). We can see that the peak height of the MTF sidelobe is only related to the number of sub-pupils of the segmented telescope, input wavelength and piston error between segments based on the coherence measurement of a star image when using broadband input light. So the amplitudes of the MTF sidelobe can be easily calculated when the piston error is known. However, on the other hand, due to the complexity and high nonlinearity of Equation (16), the piston error cannot be directly solved when the peak height of the MTF sidelobe is measured. 

### 2.2. Verification of the Correctness of the Theoretical Relation by MATLAB Simulation 

In order to verify the correctness of the mathematical derivation for Equation (16), we set up an optical system in MATLAB and simulated the corresponding MTF using the MATLAB program. The simulation segmented telescope system we used here is composed of two sub-pupils with mask, as shown as Figure 1. The left sub-mirror is the reference pupil; the piston error was introduced on the right pupil. The sampling grid of the exit pupil plane was set as 256 × 256 pixels, the pixel size of the CCD was 3.5 μm and the F^#^ of the optical system was 8. Thus the circumscribed circle diameter of the single hexagonal sub-mirror was 59 pixels, the diameter of the circle on the mask was 18 pixels and the distance between the centers of the two circles was 52 pixels to satisfy the Nyquist sampling criterion. The central wavelength of the input broad light was 632.8 nm and its bandwidth was 1 nm. The effective detection range of the proposed method is half of the coherent length:(17)$$L=\frac{L_{c}}{2}=\frac{\lambda_{0}{}^{2}}{2\Delta \lambda }=\frac{{\left(632.8\;nm\right)}^{2}}{2\times 1\;nm}\approx 200\;\mu m,$$
where $L_{c}$ is the coherent length of the input light, the factor 1/2 is due to the input light reflection on the segmented primary mirror giving an OPD equal to twice the step between segments.

Based on the set system model, different piston errors were introduced to the right sub-pupil. We first obtained the system MTF using the MATLAB program, which directly performs 2D discrete Fourier transform for the simulated CCD captured image of the point-source object. Then by taking the corresponding piston errors to Equation (16), we got the theoretical values of $MTF_{sidelobe-broad-peak}$. By comparing the theoretical calculated values from Equation (16) with the MATLAB simulation results, we verified the correctness of the theoretical formula established based on the Fourier optics principle.

Figure 2 shows four sets of MTF obtained by the MTALAB program and the peak height value of the MTF sidelobe is marked in the corresponding figure, where the introduced piston errors were 0 μm, 50 μm, 100 μm and 200 μm, respectively. We took the four sets of piston errors into Equation (16), the *n* was set to 100 in order to make the differential summation closer to the integral, and N=2. Then the calculated corresponding peak heights of the MTF sidelobe were 0.5, 0.4503, 0.3187 and 6.0708 × 10^−4^, respectively. We can see that the surrounding peaks of the MTF calculated by Equation (16) were the same as the results directly obtained from the MATLAB program.

In order to further verify the correctness of Equation (16), 201 sets of piston errors were introduced in 1 μm steps from 0 to 200 μm. The peak heights of the MTF sidelobes calculated from Equation (16) and the MATLAB program are shown in Figure 3a,b. Figure 3 verifies the correctness of the theoretical relationship between the peak heights of the MTF sidelobes and the piston error shown as Equation (16). When the MTF was obtained, the piston error should have been solved by Equation (16) theoretically. However, due to the nonlinearity and complexity of Equation (16), it is difficult to calculate the piston error in a realistic segmented telescope directly.

Note that Figure 3 only shows the positive side of the piston error. If two piston errors have the same absolute value but opposite signs, they will have the same peak heights of MTF sidelobes due to the fact that Equation (16) is a symmetrical function. Namely, we need to know the relative spatial position between the sub-mirror to be measured and the reference sub-mirror in advance in order to obtain the correct piston error. This is one of the inconveniences when using our proposed algorithm.

## 3. Piston Error Detection Method Using BP Artificial Neural Network

### 3.1. Brief Introduction of the BP Artificial Neural Network

Equation (16) directly gives the theoretical relationship between the piston error and the peak heights of the MTF sidelobes. However, due to the high nonlinearity and complexity of Equation (16), it is difficult to calculate the piston error in a realistic segmented telescope when the MTF is obtained. Thus, we turned to machine learning and neural networks [27] to establish the mapping relationship between the MTF’s surrounding peak heights and the piston error rather than fitting Equation (16) with precise mathematical expressions.

Artificial neural networks [28,29,30], which belong to machine learning, are input–output information processors composed of parallel layers of elements or neurons, loosely modeled on biological neurons, which possess local memory and are capable of elementary arithmetic. They can be used to learn and store a great deal of nonlinear mapping relations from the input–output model. The artificial neural network, composed of many neurons, has a parallel distribution structure. Each neuron has a single output and can be connected to other neurons; there are multiple output connection methods, and each connection method corresponds to a connection weight coefficient. The artificial neural network can be regarded as a directed graph formed by connecting weighted directed chains with processing elements as nodes. For the *k*th neuron, if the signal from another neuron is $x_{i}$, their interaction weight is $w_{ik}$ and the internal threshold of the processing unit is $\theta_{k}$, then the input of the *k*th neuron is $\sum_{i=1}^{m}x_{i}w_{ik}$, where *m* is the number of the input neurons connected to the *k*th neuron. Thus, the output of the *k*th neuron $y_{k}$ is:(18)$$y_{k}=f\left(\sum_{i=1}^{m}x_{i}w_{ik}-\theta_{k}\right),$$
where f() is called the activation (or transfer) function. The purposes of the activation function are introducing nonlinearity to neural networks and bounding the value of the neuron so that the neural network is not paralyzed by divergent neurons. A common example of activation function is the sigmoid (or logistic) function, which is shown as:(19)$$\phi (z)=\frac{1}{1+\exp(-z)}.$$

When the neural network structure is designed and has inputs and outputs, the network needs to be trained so it can learn the appropriate mapping relationships between the inputs and the outputs. The BP algorithm is one of the most widely used algorithms for training neural networks, and utilizes a gradient descent method to minimize the square of the error between the network output value and the target value to adjust the weights of each neuron. The BP algorithm can be divided into two steps: forward propagation and back propagation. The forward propagation process is propagating the input signals from the input layer, via the hidden layer, to the output layer to get the final output results. The back-propagation process is propagating the error signal, which is the difference between the real output and the expected output of the network, from the output end to the input layer in a layer-by-layer manner to adjust the weights and thresholds. Through one forward propagation and one back propagation, we can update the parameters of the network once. The network training process is to constantly reciprocate the forward and back propagation and continuously update the network parameters, and finally make the network precisely map the real relationships between the inputs and outputs. A sketch graph of a BP artificial neural network with two hidden layers is shown in Figure 4.

### 3.2. Piston Error Detection Approach with BP Artificial Neural Network

The piston error detection for multiple sub-mirrors of a segmented telescope with a BP artificial neural network can be divided into three steps:
(1)Determine the system parameters and generate the datasets for training the neural network under the specified system parameters. From Equation (16), we can see that the peak heights of the MTF sidelobes are only related to the input broadband light, the number of sub-pupils and the piston errors between the multiple sub-mirrors. When the segmented telescope and its working wavelength are determined, within the half of the coherent length of input light, a set of piston errors between segments is randomly introduced, and by taking the corresponding piston errors into Equation (16), the peak heights of MTF sidelobes can be obtained. The peak heights of MTF sidelobes directly calculated from Equation (16) served as one column of input matrix and the corresponding piston errors served as one column of the output matrix. Thus the input dataset and output dataset of the network could be generated.(2)Establish the neural network and train it with the input dataset and the corresponding output dataset. Here we utilized the neural network fitting tool in MATLAB, and by properly setting the number of neurons in each layer, the neural network was established. In the training process, the dataset is separated into three parts including training set, validation set and test set, then a specific training algorithm is set to train the network.(3)Once the network is well trained, we can apply the trained neural network to determine the piston errors with the PSF image collected from the optical system. Note that the image should be Fourier transformed first to get the peak heights of the MTF sidelobes before they can be handled with the neural network. By establishing the correspondence of the MTF sidelobes with their associated sub-pupils, multiple piston error measurements of the whole aperture can be implemented simultaneously by one detection of a CCD broadband image.

The application procedure of the piston error detection approach with BP artificial neural network is summarized in Figure 5.

## 4. Simulation

### 4.1. Piston Error Detection for the Telescope Composed of Two Segments

Here we first utilized the telescope model composed of two hexagonal segments in Figure 1 to test the proposed method. According to Equation (16), the peak height of the MTF sidelobe is only related to the number of sub-pupils, input wavelength and piston error. Here the number of sub-pupils was N=2, the light source was broadband light centered at 632.8 nm with bandwidth 1 nm, the left sub-mirror was set as the reference pupil, and a series of piston errors were introduced to the right pupil to generate the corresponding peak heights of MTF sidelobes. Considering the coherence length, 100,000 sets of piston errors were randomly generated between 0 μm and 200 μm to produce a corresponding 100,000 sets of surrounding peak heights of the MTF based on Equation (16). Thus, we could obtain the dataset for training the network. Then the neural network could easily be trained using the neural network fitting tool in MATLAB (the number of nodes in the hidden layer was set as 50), the peak heights of the MTF sidelobes were fed into the network and the corresponding piston errors constituted the output dataset. In this process, the dataset was separated into three parts; i.e., training set, validation set and test set. The training set was used for learning, which was to fit the weights of the network; the validation set was used for tuning the final architecture of the network; the test set was only used for assessing the performance of the network. The ratio between them in this work was 70%: 15%: 15% (i.e., 70,000 sets for training, 15,000 sets for validation and 15,000 sets for testing).

The training results are shown in Figure 6, which provides the distribution of the error between the targets and the actual outputs of the network in the form of a histogram. From the results, we can see that for the vast majority of the cases the errors between the targets and outputs were within 0.01 μm, which equals 1.5% of the input central wavelength. More specifically, the root mean square errors (RMSE) between the targets and outputs of the network in the training set, validation set and testing set were 2.531 × 10^−5^ μm, 1.914 × 10^−5^ μm and 2.465 × 10^−5^ μm, respectively. This demonstrates the high accuracy of the neural network with the peak heights of the MTF sidelobes serving as the input to solve the piston error.

Then we apply the obtained neural network to the PSF images collected from the established optical system composed of two hexagonal segments in MATLAB described in Section 3.2. To approximate the practical imaging environment, a zero mean and 0.05 variance Gaussian distribution noise were introduced in the simulated PSF images. A Fourier transform was performed for the noised PSF image to obtain the OTF, which was composed of the MTF and phase transfer function (PTF), where the peak heights of the MTF sidelobes served as the input. The well-trained neural network could directly output the piston error introduced on the right sub-mirror corresponding to each PSF image; we named the actual output of the network as the measured piston error. The difference between the measured piston errors and the settled piston errors is shown in Figure 7. Seventeen sets of experimental results are listed in this figure. From the error analysis, the RMSE of all the difference was 1.3 nm. Here the RMSE was calculated from the following formula:(20)$$RMSE=\sqrt{\frac{1}{N}\sum_{n=1}^{N}Difference_{n}^{2}},$$
where *N* is the total number of the datasets.

From the simulation results, we can see that the piston error detection method based on a BP artificial neural network had a quite high accuracy and large capture range. However, when approaching the two ends of the capture range, the detection accuracy decreased slightly. This was due to the peak heights of the MTF sidelobes approaching the edges of the capture range not changing with piston error very obviously; namely the gradients of the curve were small in these parts, and we can see this directly from the theoretical relationship curve between the piston errors and peak heights of MTF sidelobes shown in Figure 3a. This results from the insensitiveness of the network to data approaching the ends of the capture range, so the detection accuracy decreased a little bit, but the accuracy at the two ends of the detection range was still high enough to meet the requirement of co-phasing the piston error.

We also compared our proposed method with the Gaussian fitting and two-piece quartic polynomial function fitting methods presented in [21,22], respectively, and the comparison results on detection accuracy are shown in Figure 8. From Figure 8a, we can see that the detection accuracy of our proposed method is much higher than that of the Gaussian fitting method. Figure 8b shows the detection accuracy curve of our method compared with that of the two-piece quartic polynomial function fitting method. Our method produced larger differences (~5 nm) for small piston errors, but with the increase of the piston error, especially after 125 μm, the two-piece quartic polynomial function fitting method generated much larger differences. From the error analysis, the difference of our method was 1.3 nm RMS, which is smaller than the 1.8 nm RMS presented in [22]. This demonstrates that our method also has higher detection accuracy than the two-piece quartic polynomial function fitting method, while the implementation complexity is lower since there is no need to build different computation models in different intervals of the capture range.

To further evaluate the ability of the trained network, we designed some other simulations to discuss the effect of image noise and the tip-tilt aberrations of the segments on the detection accuracy of the proposed method. To approximate the practical imaging environment, different intensity Gaussian distribution noises were introduced in the simulated PSF images. Figure 9a gives the detection accuracy under three different image noise intensities by the trained network, and the simulation result showed that PSF image noise had little effect on the piston error detection accuracy. This was due to the fact that the Gaussian distribution noise contributed little to the peak heights of MTF sidelobes, so the proposed piston error detection algorithm had a very good noise immunity. Additionally, since the tip-tilt aberrations could not be corrected entirely before measuring the piston error, slight tip-tilt distortions were introduced to the optics system while generating the simulated PSF images. Here we introduced two different tip-tilt aberrations: the RMS errors were 0.01λ and 0.05λ (λ=632.8 nm), respectively. Since the introduced tip-tilt aberrations were quite small, we assumed that the aberration distribution for each wavelength was the same. The effect on the piston error detection accuracy is shown in Figure 9b, which confirms that the existence of the tip-tilt error had a great influence on the detection accuracy. This was because that the tip-tilt error greatly changed the amplitudes of MTF sidelobes. So, in order to guarantee the detection accuracy, the tip-tilt aberrations should be corrected well enough (at least less than 0.01λ RMS) before piston error measurement.

From Equation (16), we can conclude that the datasets used to train the network are only related to the numbers of telescope sub-pupils and input light sources. When the numbers of the sub-pupils and working wavelength were determined, the network could be trained to map the relation between the peak heights of MTF sidelobes and piston error, since learning mapping relationships is a data-driven process. Thus, even for different imaging systems, if they have the same numbers of sub-pupils and work under the same operating light, the trained network should have quite good generalization ability. In order to verify this, we designed another simulation based on the previous one. In this simulation, we changed the F^#^ of the imaging system and generated different intensity images from these systems as testing sets to examine the generalization ability of the trained network (original imaging system parameters are given in Section 3.2 with $F^{\#}=8$) for the new systems. The F^#^ of the imaging systems used for simulations were 10, 20, 30, 40, 50 respectively, while the piston error measurement results of all the new systems using the trained network had the same accuracy as those of the original imaging system with $F^{\#}=8$. This demonstrates that our network has quite good generalization ability.

### 4.2. Simultaneous Multi-Piston Measurement

In the part, we utilized the proposed method to simultaneously detect multi-piston errors of the whole aperture. Based on the theoretical deduction in Section 3.1, we can see that the MTF model of the *N* sub-pupils segmented telescope consisted of *N*^2^ sub-MTFs. In the spatial frequency domain, the *N* sub-MTFs overlapped at the position where the center spatial frequency was zero to form the central peak, while the other *N* (*N* − 1) sub-MTFs distributed around the central peak to form the sidelobes. Every pair of sub-pupils produced a pair of MTF sidelobes, the sidelobes symmetrically distributed on both sides of the central peak. If all of the sidelobes did not overlap, their amplitudes could be obtained at the same time by one CCD image, hence the piston errors of all sub-mirrors were retrieved at the same time by inputting the peak height of sub-MTF corresponding to each sub-mirror into the trained network.

In order to verify the feasibility of simultaneous multi-piston measurement with the proposed method, we took a primary mirror composed of four hexagonal sub-pupils as an example. We modeled the four-segment imaging system using MATLAB, results of which are shown in Figure 10a. The No. 1 sub-mirror was set as reference pupil, and the piston errors were introduced on the No. 2, No. 3 and No.4 sub-pupils, respectively. The corresponding system MTF without piston errors is shown in Figure 10b: there was one MTF central peak and 12 MTF sidelobes (*N* (*N* − 1) = 4 × (4 − 1) = 12). Four sub-MTFs (*N* = 4) overlapped at the position where the center spatial frequency was zero to form this central peak, and every pair of sub-pupils produced a pair of MTF sidelobes. Note that a colored MTF was used here in order to describe which peaks were actually used to solve the piston error of the corresponding sub-mirror. When the No. 1 sub-mirror was set as reference pupil, No. 2 sub-mirror produced the two red sub-MTFs, No. 3 sub-mirror produced the two green sub-MTFs and No. 4 sub-mirror produced the two yellow sub-MTFs. The six light blue sub-MTFs at the outermost periphery were produced either by No. 2 and No. 3 sub-mirrors together, or by No. 2 and No. 4 sub-mirrors together, or by No. 3 and No. 4 sub-mirrors together, which could not be used to measure piston errors. Thus, we could use any one of the red sub-MTFs to measure the piston error of No. 2 sub-mirror, any one of the green sub-MTFs to measure the piston error of No. 3 sub-mirror and any one of the yellow sub-MTFs to measure the piston error of No. 4 sub-mirror, respectively. For the detailed generation principle between the sub-MTFs and sub-mirrors, refer to paper [31].

In order to simultaneously detect piston errors of the three segments, we first had to establish the neural network and train it with the generated dataset. Here we still used the broadband light source centered at 632.8 nm with bandwidth 1 nm, since *N* = 4, and according to Equation (16) we could directly obtain the datasets of the peak heights of the MTF sidelobes varying with the piston errors to train the network. The curve of the relation between the peak heights of the sub-MTF and the piston errors is shown in Figure 11. One hundred thousand sets of data were fed into the network to train it, and the network settings were the same as those of the two sub-pupils’ telescope except the ratio between the training set, validation set and test set was 65%: 20%: 15%. We added more data to the validation set to avoid the overfitting problem. The training results are shown in Figure 12.

Then we used the PSF images directly collected from the established optical system composed of 4 hexagonal segments shown in Figure 9a to simultaneously measure piston errors for all sub-mirrors based on the trained network. We randomly introduced different piston errors between 0 μm and 200 μm to No. 2, No. 3 and No. 4 sub-mirrors at the same time, then performed Fourier transform for the obtained PSF image to get the system MTF. Here we also added zero mean and 0.05 variance Gaussian distribution noise to the simulated PSF images. According to the correspondence relation between the MTF sidelobes and sub-mirrors (here No. 4 sub-MTF corresponded to No. 2 sub-mirror, No. 6 sub-MTF to the No. 3 sub-mirror and No. 5 sub-MTF to No. 4 sub-mirror, respectively), by inputting the peak height of sub-MTF corresponding to each sub-mirror into the trained network, the piston errors of all the sub-mirrors could be measured by one CCD image at one time. A part of the simulation results is listed in Figure 13. The piston error measurement results of No. 2, No. 3 and No.4 sub-mirrors are given by Figure 13a–c, respectively. Figure 13d shows the total RMSE of the three sub-mirrors’ piston error measurement accuracy during each test.

From the simulation results we can see that the multi-piston errors between the segments of the entire aperture could be simultaneously detected with high accuracy: the average value of the RMS errors over the testing samples was about 1.4 nm. Since the peak height of sub-MTF corresponding to each sub-mirror was fed into the trained network to measure the piston errors, the most important issue was to avoid the MTF sidelobes respectively produced by the reference sub-pupil and all of the measured sub-pupils overlapping. For the detailed arrangement rules, refer to paper [31] to confirm the MTF sidelobe distribution was non-redundant.

Finally, we tried to compare our work with Ma’s work presented in paper [32], since this also used a single broadband image to sense the piston errors between sub-pupils. Rather than establishing the theoretical relationship between the MTF sidelobes of PSF images and the piston errors like us, this research directly used in-focused broadband images as the input, and established one deep convolutional neural network (DCNN) to learn to sense pistons with a single broadband focal image. The simulation results for the two-pupil imaging system and four-pupil imaging system are shown in Figure 14.

From the simulation results, we can see that our method has higher piston error measurement accuracy than Ma’s method presented in paper [32]. This improvement of the detection accuracy is at the expense of extensive complex optical calculations in advance in our work, since we used the MTF sidelobes of PSF images as inputs for the network based on the established theoretical relationship between the MTF sidelobes and the piston errors, while Ma directly used in-focused broadband images as the input.

## 5. Conclusions

In this paper, we put forward a method to simultaneously detect the multi-piston errors between the segments based on the broadband intensity distribution on the image plane by a BP artificial neural network. A mask with a sparse sub-pupil configuration was set on the exit-pupil plane to sample the wave from the segments. Based on the Fourier optics principle, the accurate theoretical relation between peak heights of MTF sidelobes and piston errors was obtained. Instead of fitting the relation by Gaussian function or two piecewise quartic polynomials, we utilized a BP artificial neural network to establish the nonlinear mapping relationship between the piston errors and the amplitudes of the MTF’s surrounding peaks. By introducing different piston errors into the segmented optics system, the corresponding peak heights of MTF sidelobes could be directly calculated by the established theoretical formula, while the corresponding amplitudes of MTF sidelobes and the introduced piston errors were used as the inputs and outputs, respectively, to train the network. Once well trained, it could estimate the piston error to a good precision with high efficiency and robustness using the PSF images collected from the optics imaging system as inputs. Since the MTF sidelobes of all the sub-pupils can be simultaneously obtained by one detection of a CCD image, multiple piston errors of the entire aperture can be retrieved at one time.

Adequate simulation experiments were implemented to demonstrate the effectiveness and accuracy of the proposed approach. We established the models of a segmented telescope composed of two hexagonal segments and four hexagonal segments successively in MATLAB and implemented the PSF images collected from these simulated optical systems to test the performance of the corresponding trained networks, respectively. The piston sensing simulation results showed that the average values of RMS errors on the two-segment imaging system and the four-segment imaging system could achieve 1.3 nm and 1.4 nm, respectively, and the method’s capture range was the operating light’s coherence length. Compared to Gaussian fitting and two-piece quartic polynomial function fitting methods, our method has higher detection accuracy and is easier to implement. The influence of the CCD camera noise and the existence of tip-tilt aberrations on the accuracy of piston error detection were also discussed. The results indicated that the proposed piston error detection algorithm has a very good noise immunity, but the tip-tilt aberration should be corrected well enough before the piston error measurements. Besides, different intensity images from the imaging system with different F^#^ values were generated and fed into the original trained network to test the network generalization ability. The piston error detection accuracy was the same as that of the original imaging system, which demonstrates that our network has quite good generalization ability.

The hardware cost of our method is quite small; only a mask with a sparse multi-subaperture configuration is needed to attach in the exit pupil plane of the segmented telescope, and this mask should ensure the MTF sidelobes’ non-redundant distribution. Thus, with this method, multi-piston measurements of the whole aperture can be implemented simultaneously, and piston detection no longer need be divided into coarse and fine regimes, which involves separate dedicated hardware solutions. In view of the efficiency and superiority, it is expected that the piston sensing method based on the BP artificial neural network proposed in this paper can be adapted to any segmented and deployable primary mirror telescope, no matter the shape of the segmented mirror and the number of the segments.

## Figures and Tables

**Figure 1 sensors-21-03364-f001:**
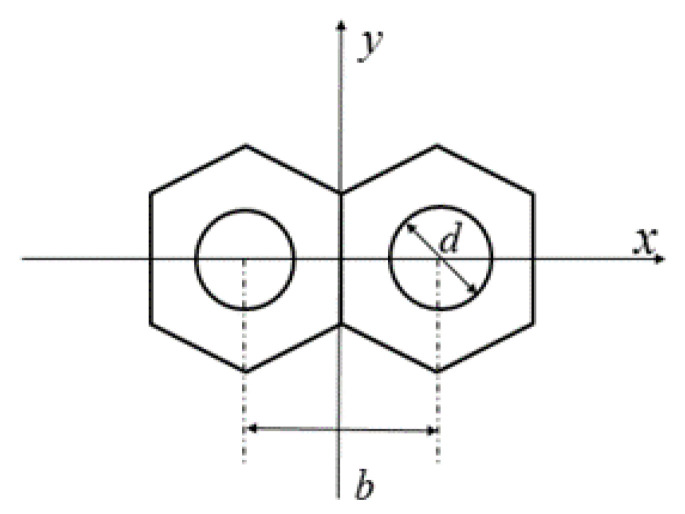
The schematic diagram of the telescope composed of two hexagonal segments with sparse circles configuration.

**Figure 2 sensors-21-03364-f002:**
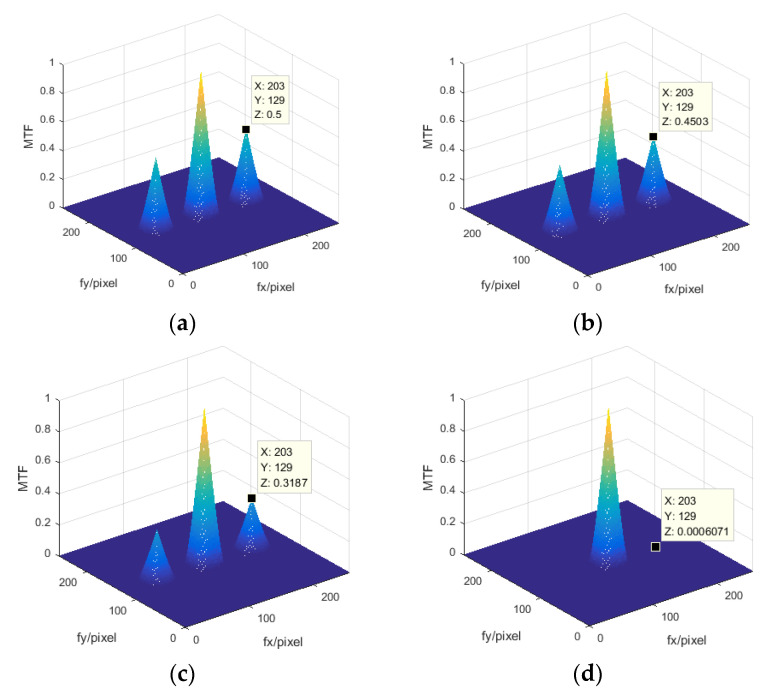
Modulation transfer function (MTF) of four different piston errors obtained by MATLAB simulation. (**a**) Piston error = 0 μm, (**b**) piston error = 50 μm, (**c**) piston error = 100 μm, (**d**) piston error = 200 μm.

**Figure 3 sensors-21-03364-f003:**
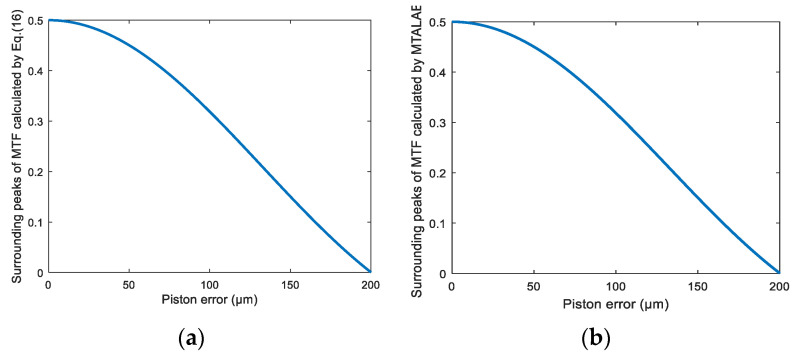
Surrounding peak heights of MTF varying with the piston error (**a**) calculated by Equation (16) and (**b**) obtained by the MATLAB program.

**Figure 4 sensors-21-03364-f004:**
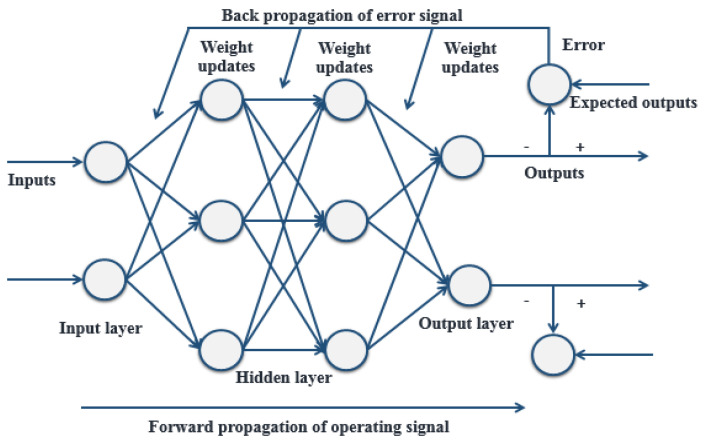
Sketch graph of a back-propagation (BP) artificial neural network with two hidden layers.

**Figure 5 sensors-21-03364-f005:**
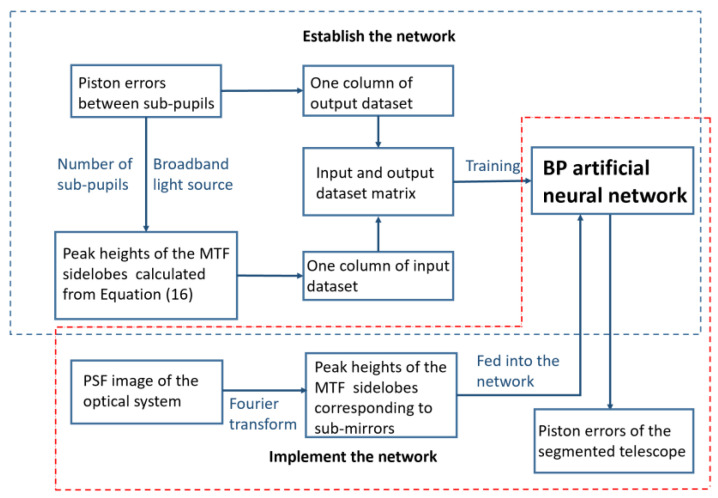
The detailed piston error detection approach with a BP artificial neural network.

**Figure 6 sensors-21-03364-f006:**
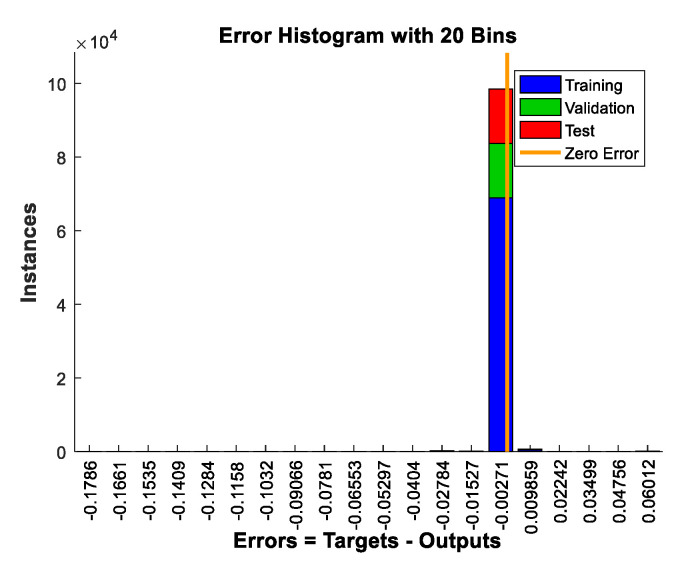
Distribution of the errors between the targets and the actual outputs of the network in the form of a histogram.

**Figure 7 sensors-21-03364-f007:**
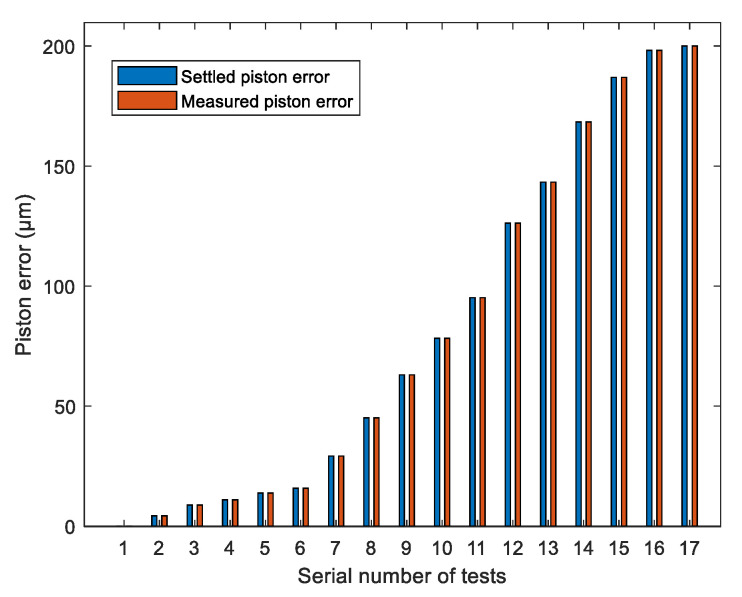
The piston error detection results in bar chart form for two segments system with a BP artificial neural network.

**Figure 8 sensors-21-03364-f008:**
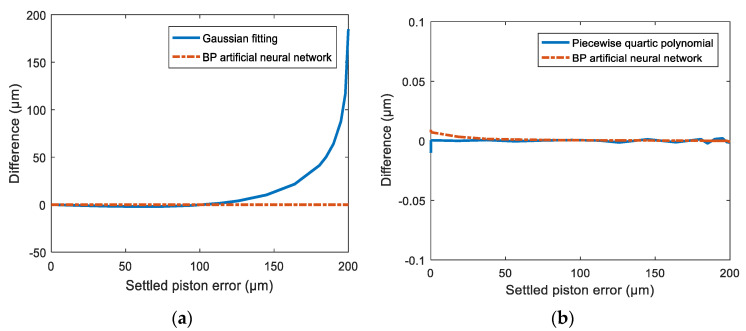
The comparison results on detection accuracy of a BP artificial neural network approach with (**a**) the Gaussian fitting method (**b**) and the two-piece quartic polynomial function fitting method.

**Figure 9 sensors-21-03364-f009:**
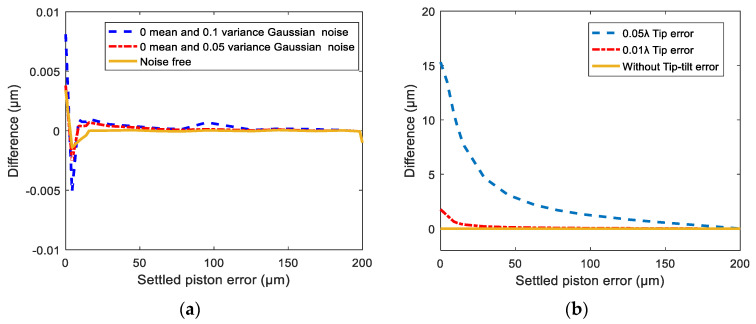
The piston error detection accuracy under (**a**) different CCD image noise values and (**b**) different tip-tilt aberrations.

**Figure 10 sensors-21-03364-f010:**
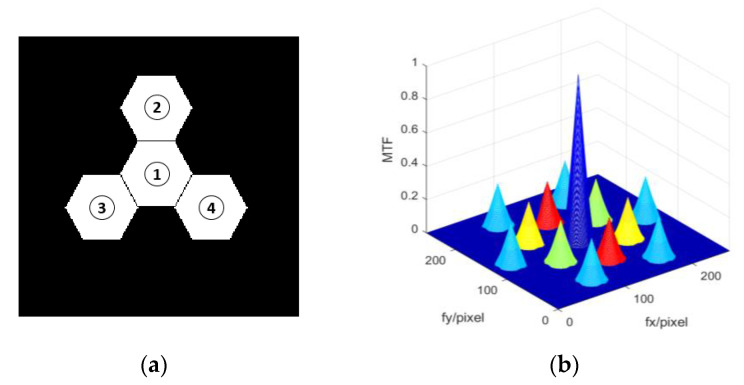
The 4 hexagonal segmented telescope showing (**a**) the position arrangement of the 4 sub-mirrors and (**b**) the corresponding system MTF colored without piston error.

**Figure 11 sensors-21-03364-f011:**
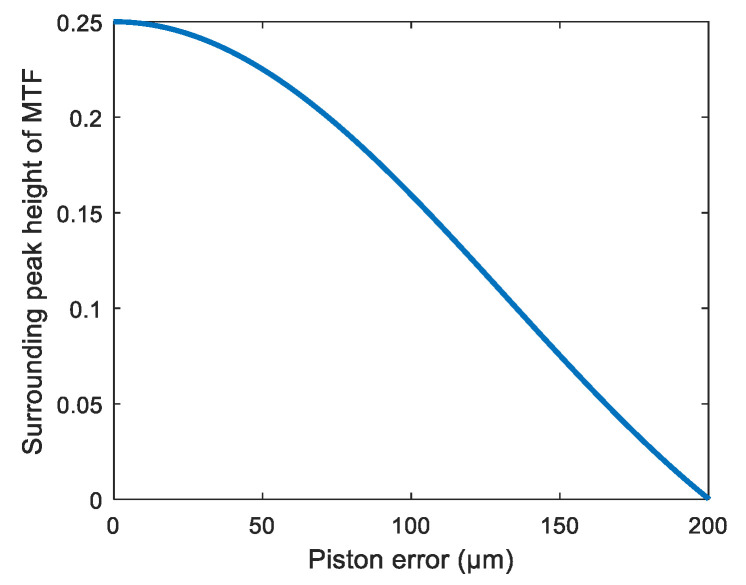
Peak heights of the MTF sidelobes varying with the piston errors of the optics system composed of 4 sub-mirrors.

**Figure 12 sensors-21-03364-f012:**
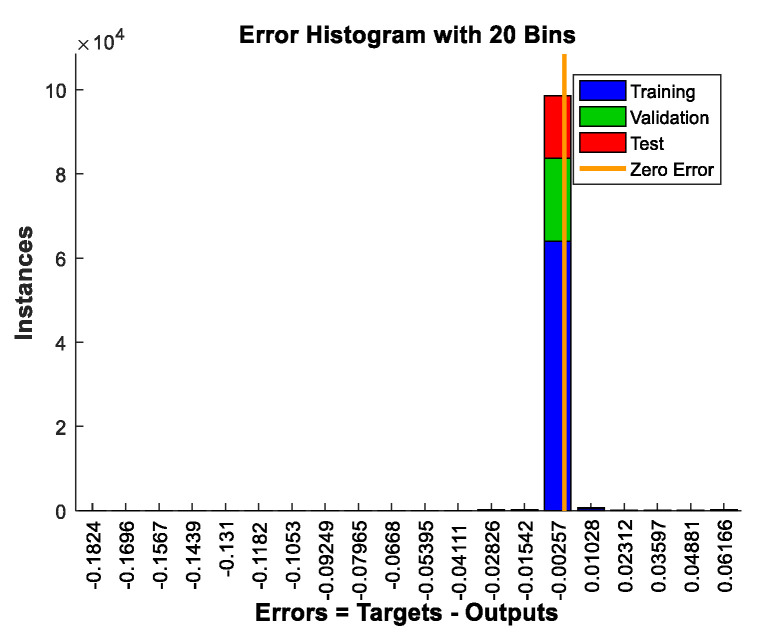
The network training result for the optics system composed of 4 sub-mirrors.

**Figure 13 sensors-21-03364-f013:**
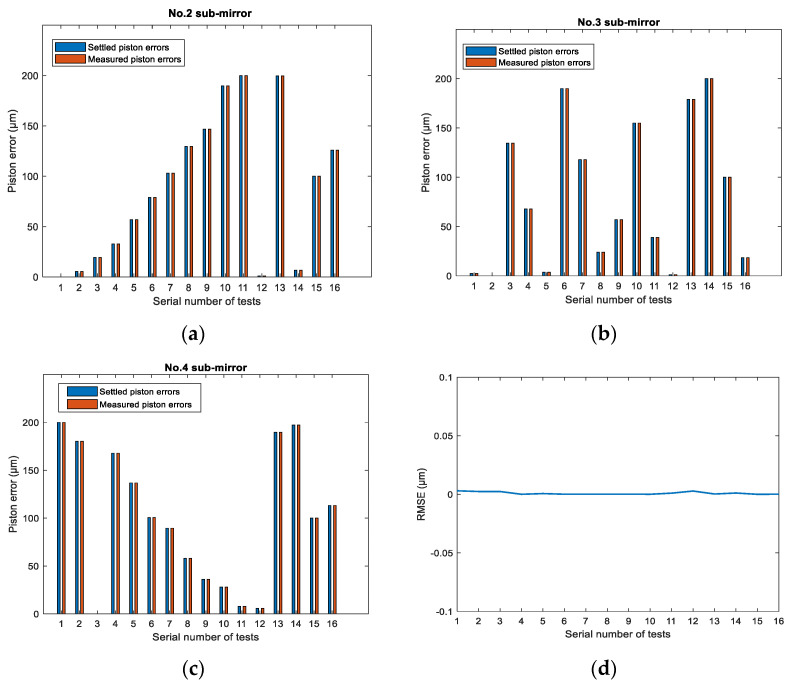
Piston error measurement results for the four-segment system: (**a**–**c**) piston error measurement results of No. 2, No. 3 and No.4 sub-mirrors, respectively, and (**d**) the total RMSE of the three sub-mirrors’ piston error measurement accuracy during each test.

**Figure 14 sensors-21-03364-f014:**
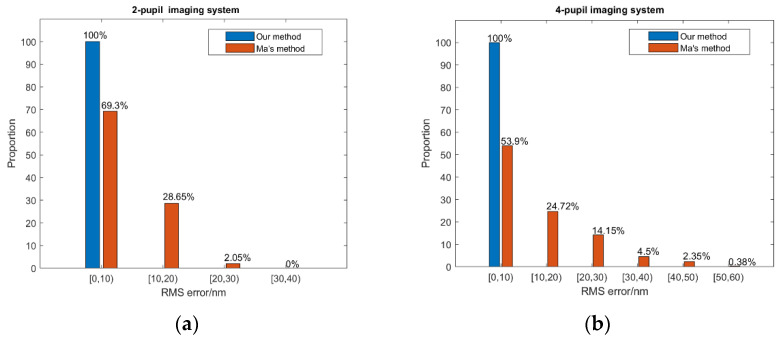
The piston error detection accuracy comparison between our method and Ma’s method (**a**) for the 2-pupil imaging system and (**b**) the 4-pupil imaging system.

## Data Availability

Not applicable.

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
