# Peer review of "Piston Error Measurement for Segmented Telescopes with an Artificial Neural Network"

_sensors, 2021, doi:10.3390/s21103364_

Round 1

Reviewer 1 Report

Nice work. There are some issues which must be taken care of:

  1. The MTF sidelobes are very wavelength-elongated, but claimed to be narrow.
  2. The piston error can be either positive or negative, but the sidelobe height cannot tell them apart. It's not clear how the neural network deals with this.
  3. A similar experiment by Paykin et al. with real data (but using manual methods) showed the above issues. The reference is Physical Review E 91, 023302 (2015).
  4. Typos and other comments are in the attached file.

Reviewer 2 Report

Dear Authors,

  congratulations to your interesting work, which I think merits quick publication. A couple of remarks I do have which can or cannot be addressed at your discretion, but I think taking them into account would improve the paper.

p1, l 33-35: E-ELT (now just ELT) and TMT do not yet exist...

p1, bottom: Strictly speaking, the Zernike basis cointains only 1 piston term, defined across the full aperture. For segmented apertures, individual segment pistons either show up as very high order Zernike terms, or one needs to define the basis for each segment individually.

General remark: For the ELT, the piston error between segments is not a concern as reference surfaces and sensors are expected to mitigate this problem to a satisfying accuray of about 10nm. Fragment piston however, piston errors between the sub-apertures divided by the secondary support structure (aka "spiders") are a major cause of concern, since those fragments cannot be easily aligned, or their piston error sensed.

p11, Sec3.: It would be nice to see some MATLAB code actually implementing the chosen BP NN ...

p13, Fig.7b: The figure apparently indicates tahht PQP provides a better determination of piston erros than BP NN., while the text says the opposite.  How are the 1.3 and 1.8nm rms determined that you are talking about? In particular for small piston errors, BP NNs seem to produce much larger  differences (~5nm) according to Fig 7b.

p14. l 440ff:  tip tilt is a tricky business, related to the general remark above: A tilt across the full aperture is in fact a piston error between segments/fragments, and vice versa. Simply imagine the rightmost segment/fragment sitting a bit higher than the leftmost segment/fragment, simply because the wavefront is tilted.  On the other hand, a step-wise ever increasing piston is an approximation, and thus a cause for real tilt.  So what you are discussing here is in fact not a problem, but more property of the wavefront nature.  If you are talking about the 984 segments of the ELT, even higher-order static aberrations will introduce piston offsets between segments.  These (and tip-tilt) will and must be corrected by something different than piston sensing, though.

  On the other hand, measurements will usually be taken on-sky, while closed-loop adaptive optics correction is in operation.  So instead of static errors, you will be faced with dynamic errors, and they will be much larger than 0.01lambda.  The typical TT residual at the ELT in closed-loop AO operation is expected to be 1mas, which corresponds to a tilt-induced piston difference between opposing outer edge segments of about 200nm.

So it would be interesting to have simulation where you add typical AO residual wavefront aberrations on top of your piston pattern, to see how this influences the BP NN's capability to reconstruct piston.  Of course it shouldnt be static, but average over a couple of independent realizations instead, as many as correlation times (~30ms) fit into an expected integration time.

p15, l488ff: The pattern starts to get complex and the written description too. Would it be possible to think of an adeqaute colour scheme to improve Fig 9b in order to demonstrate which peaks are atually usable?

Again, thanks for and congratulations to the nice paper - and I'd be most interested to see the method applied to the 6 large ELT fragments under closed-loop on-sky conditions!

Reviewer 3 Report

  1. This article is full of simulations and lacks real data. We recommend that you supplement real data to experiment.
  2. It is a good idea to use neural network to detect errors in this paper. However, when using BP, the author did not make corresponding improvement of neural network, and the innovation is limited.
  3. In this article, how is the data set of the BP neural network generated? How does the author judge the advantages of the method proposed in the article?
  4. In this article, we only see the author comparing with proposed method. Are there other advanced methods in the field of Piston error measurement? The method proposed by the author lacks comparison with existing methods.
  5. The calculations of the formulas are listed extensively in the article, but the meaning behind these formulas and the corresponding principles are not explained.

Round 2

Reviewer 1 Report

Point 4: This graph (Fig. 3) only shows the positive side of the error, but the experiment cannot tell if the piston is positive or negative. The fringe contrast depends on the cosine of the difference, which is a symmetrical function.

Response 4: Based on your advice, we checked both our theoretical formula derivation and software simulation, we found out that when the absolute value of the piston error is the same, the corresponding MTF sidelobes are the same. It is true that the experiment cannot tell if the piston is positive or negative, not only for our work but also for JUNLUN JIANG’s work in paper [22]. The calculated piston error is more like the absolute value of the two sub-mirrors relative position. We will try to improve this in our further work.  

Point 4 (repeat): Please include the above answer in your final version, otherwise it will be misleading. If Mr. Junlun Jiang was wrong, it does not make sense to repeat his error.

Points 5 and 6: Indeed the graphics in the new version are clearer. You should consider dropping the two tables altogether.

Reviewer 3 Report

Accept.

Author Response

Dear Reviewer:

Thank you very much for your acceptance of our response to your professional and detailed review comments. We are also grateful for your time and effort spent on our article. 

Best Regards,

Dan Yue

This manuscript is a resubmission of an earlier submission. The following is a list of the peer review reports and author responses from that submission.